# Molecular Insights into the Wall Slip Behavior of Pseudoplastic Polymer Melt in Nanochannels during Micro Injection Molding

**DOI:** 10.3390/polym14153218

**Published:** 2022-08-08

**Authors:** Wangqing Wu, Fengnan Duan, Baishun Zhao, Yuanbao Qiang, Mingyong Zhou, Bingyan Jiang

**Affiliations:** State Key Laboratory of High Performance Complex Manufacturing, School of Mechanical and Electrical Engineering, Central South University, Changsha 410083, China

**Keywords:** wall slip, poiseuille flow, united-atom model, polymer melt, micro injection molding

## Abstract

Wall slip directly affects the molding quality of plastic parts by influencing the stability of the filling flow field during micro injection molding. The accurate modeling of wall slip in nanochannels has been a great challenge for pseudoplastic polymer melts. Here, an effective modeling method for polymer melt flow in nanochannels based on united-atom molecular dynamics simulations is presented. The effects of driving forces and wall–fluid interactions on the behavior of polyethylene melt under Poiseuille flow conditions were investigated by characterizing the slip velocity, dynamics information of the flow process, and spatial configuration parameters of molecular chains. The results indicated that the united-atom molecular dynamics model could better describe the pseudoplastic behavior in nanochannels than the commonly used finitely extensible nonlinear elastic (FENE) model. It was found that the slip velocity could be increased with increasing driving force and show completely opposite variation trends under different orders of magnitude of the wall–fluid interactions. The influence mechanism was interpreted by the density distribution and molecular chain structure parameters, including disentanglement and orientation, which also coincides with the change in the radius of gyration.

## 1. Introduction

As a crucial solution for implementing large-scale polymeric micro-component production, micro injection molding [1,2,3] has become the primary processing method for microplastic components in the aerospace, biomedical, precision optics, and other fields [4,5]. The micro injection molding process shows various flow phenomena that are noticeably distinct from those of regular injection molding, such as wall slip, viscous dissipation, and so on, due to the presence of micro-scale effects in the flow process. The wall slip is a non-negligible flow boundary condition for polymer melt flow that significantly alters or even dominates the fluid flow process at the micro-nano scale. The rheology of the polymer melt and the stability of the flow field distribution are closely connected to wall slip during the filling process of the polymer melt for micro injection molding. The wall slip behavior may greatly reduce polymer melt viscosity [6], enhance velocity distribution homogeneity in the flow field, and promote melt flow. It has a direct impact on plastic micro parts molding quality [7]. However, the discontinuity effect and surface effect make it difficult to uncover the microscopic mechanism of wall slip in the experimental research of polymer melt flow.

Molecular dynamics simulations have proven to be an effective method for studying wall slip. At the mesoscopic atomic/molecular level, the explanations for macroscopic property changes can be determined. Couette flow [8,9,10,11] and Poiseuille flow [12,13] are the main flow forms studied in wall-slip molecular dynamics simulations. The finitely extensible nonlinear elastic (FENE) model is commonly used to study the Poiseuille flow of polymer melt [14,15]. Gratton and Slater [14] constructed a FENE molecular dynamics model to study the flow process of bound polymer chains, observed the surface packing effect and adhesion behavior of polymers and obtained the transition region from no-slip to slip dynamic transition. By studying the Poiseuille flow of polymer melts in nanochannels, Duan et al. [12] discussed the influence of driving force, wall–fluid interaction, and molecular chain length on flow mass transfer and slip length. The results showed that larger external force, weaker wall–fluid interaction, and longer molecular chains could favorably promote polymer melt wall slip. Similarly, the velocity distributions obtained from the above studies are all of a parabolic shape.

However, it is commonly known that polymer melts are pseudoplastic fluids, causing their velocity distribution to behave as a combination of parabolic-like sections and a plunger flow section with a flat flow front. The flow behavior of high/low-density polyethylene melts in a slit die channel was studied using a laser Doppler velocimeter by Münstedt et al. [16]. The velocity distribution fits this property, according to their experimental results [16,17,18]. The relevant simulation results [12,14,15,19] revealed that the velocity distribution of the FENE model could not reflect the pseudoplastic characteristics of the polymer, resulting in a certain distortion with the actual Poiseuille polymer flow process. Interactions and processes on the diverse length and temporal scales affect many physical phenomena and features of soft matter systems such as polymers and biological materials [20]. Different spatiotemporal scales and resolution levels correspond to different computer simulation models. We believe that the FENE model fails to adequately depict the pseudoplastic features of polymers due to its inadequate mesoscopic resolution.

In this paper, in order to reflect the polymer material properties and flow process more accurately, a united-atom molecular dynamics model for simulation was constructed. The velocity distribution with pseudoplastic characteristics was obtained. From the velocity distributions, we derived the variation of the slip velocity and revealed the molecular origin behind it. Specifically, the impact of two flow boundary conditions on the slip was thoroughly considered: driving force and wall–fluid interaction. The variation of the slip velocity was acquired from the pseudoplastic velocity distributions, and it was preliminarily analyzed in conjunction with the density distribution, followed by the self-diffusion coefficient of the polymer melt. Finally, by calculating the parameters of entanglement, orientation, and radius of gyration, the changes in the spatial configuration of molecular chains are characterized quantitatively.

## 2. Simulation Methodology

### 2.1. Pseudoplastic Fluid Flow Characteristics

The flow of the polymer melt in the micro-nano channel is a typical Poiseuille flow during micro injection molding. Polyethylene is commonly utilized in the manufacture of micro injection molded components in practical industrial production because of its superior physical and mechanical characteristics. The spatial arrangement of linear polyethylene molecular chain monomers with extended molecular chains is highly efficient to observe and calculate during numerical simulations, which are also regularly utilized due to their simple structure. As a result, the major focus of research has concentrated on modeling polyethylene Poiseuille flow in a nanochannel.

The most essential velocity distribution while flowing in a rectangular mold nanochannel differs from the Newtonian fluid distribution under typical conditions because the polymer melt (polyethylene) is pseudoplastic. The theoretical velocity distribution of the cross-sectional direction (*z*-direction) in the flow direction (*x*-direction) is given by:(1)$$v_{x}(z)=\frac{n}{n+1}{\left(\frac{\Delta P}{KL}\right)}^{\frac{1}{n}}\left[{\left(\frac{h}{2}\right)}^{\frac{n+1}{n}}-{\left(\frac{z}{2}\right)}^{\frac{n+1}{n}}\right]$$
where Δ*P* represents the flow pressure drop, *L* is the channel length, *K* represents the consistency index, *h* has been the flow channel height, and *z* represents the cross-sectional position. When the polymer melt flow geometry and the temperature and pressure field conditions to which it is subjected are determined, *P*, *K*, *L*, and *h* are all constant values; therefore, *v_x_*(*z*) is merely a function of the location of the flow channel center in respect to the height of the flow channel surface. However, the value of *n* (the non-Newtonian index) varies considerably for different polymer materials and is a critical factor in determining the velocity distribution in the polymer melt mold. The velocity distribution of the melt front in the flow channel is parabolic when the fluid is Newtonian (*n* = 1). In contrast, most of the polymers in the micro injection molding process are “pseudoplastic fluids” in the molten state (*n* < 1). The lower the *n* number, the flatter the velocity distribution curve at the front of the fluid, as shown in Figure 1.

In our simulation, the experimental method [17] stated in the preceding content was utilized to fit the velocity distributions produced from the simulations to explore the flow velocity, slip case, and pseudoplasticity. The power law fitting equation is applied for the velocity when there is no relative velocity of the fluid near the wall [16]:(2)$$v_{z}=v_{max}\left[1-{\left|\frac{2z}{H}\right|}^{(n+1)/n}\right]$$

The wall slip that occurs at this point must be considered when the polymer molecular chain near the wall has a velocity relative to the wall. The slip velocity is introduced to Equation (2) as follows:(3)$$v_{z}=v_{s}+v_{0}\left[1-{\left|\frac{2z}{H}\right|}^{(n+1)/n}\right]$$
(4)$$v_{0}=v_{max}-v_{s}$$
where the maximum velocity that can be attained is *v_max_*, the slip velocity is *v_s_*, *v*_0_ can be obtained from the difference between *v_max_* and *v_s_*. The channel longitudinal section position is *z*, *n* is the non-Newtonian exponent, and the channel height is *H*.

### 2.2. Interatomic Potential and Normalized Units

In our simulation, a united-atom model of polyethylene molecular chains is adopted in order to enhance computing efficiency over traditional all-atom modeling. The hydrogen atoms of CH_2_ (including CH_3_) in each monomer are united with a carbon atom to form a single atom in the polyethylene molecular chain model, after which the Dreiding potential [21] is employed to describe the interatomic interactions of the polyethylene molecular chain. In prior molecular dynamics simulation literature, the Dreiding potential [22,23] has been extensively employed and proved effective, and it is advantageous for modeling the spatial conformation of polymer molecular chains. At the same time, compared with the all-atom model and FENE model, studies have shown that the atomic structure of polymer melts or biomolecular aggregates can be effectively simulated for long time and large length scales by united-atom modeling with an efficient backmapping methodology [20]. In this regard, the current study may provide more realistic findings than prior work.

The Dreiding potential consists of bond interactions and non-bond interaction (Table 1).

Bond tension (*r*), bond angle bending (*θ*), and dihedral torsion (*ϕ*) are all included in the bond interaction energy:(5)$$E_{bond}\left(r\right)=\frac{1}{2}K_{b}{\left(r-r_{0}\right)}^{2}$$
(6)$$E_{angle}\left(\theta \right)=\frac{1}{2}K_{\theta }{\left(\theta -\theta_{0}\right)}^{2}$$
(7)$$E_{dihedral}\left(\phi \right)=\sum_{i=0}^{3}C_{i}{\left(\cos\phi \right)}^{i}$$

The bond tension, bond angle bending, and dihedral torsion stiffnesses are represented by $K_{b}$,${\;K}_{\theta }$, and $C_{i}$. The equilibrium bond lengths and angles are $r_{0}$ and $\theta_{0}$. The coefficients of multi-harmonic dihedral angle are contained in $C_{i}$. 

The non-bond interaction is the standard 12-6 Lennard-Jones (L-J) potential:(8)$$E_{non-bonding}\left(r\right)=4\varepsilon \left[{\left(\frac{\sigma }{r}\right)}^{12}-{\left(\frac{\sigma }{r}\right)}^{6}\right],r\leq r_{c}$$
where *r* represents the distance between two atoms, ε represents the depth of the potential energy, σ represents the location of zero potential energy, and $r_{c}$ represents a cut-off distance.

We chose Ni as one of the simulation materials; on the one hand, many mold coatings contain Ni, so it is more valuable to choose Ni as the simulation material; on the other hand, the specific force field parameters and values between Ni and polyethylene have been studied and applied in many pieces of literature [24], which are very mature and credible. The embed atom method (EAM) potential is the Ni-Ni interaction simulation potential. The atomic pair potential (pair) is improved by adding an electron-cloud-density-related term to it, making it more accurate than the pair potential alone. The EAM potential is written as [24]:(9)$$E(r_{i}{}_{j})=\frac{1}{2}\sum_{i=1}^{N-1}\sum_{j=i+1}^{N}\varphi (r_{i}{}_{j})+\sum_{i=1}^{N}F(\rho_{i})$$
(10)$$\rho_{i}=\sum_{j=1,j\neq i}^{N}\psi (r_{i}{}_{j})$$
where ${\varphi (r}_{ij})$ is the repulsion energy of two atoms *i*, *j* when their nuclei are paired, and *F* denotes the energy embed in the local electron density $\rho_{i}$ of an atom *i* at the atomic site. The local electron density is expressed by the superposition of neighboring atoms contributions ${\psi (r}_{ij})$.

It’s worth noting that the 12-6 Lennard-Jones (L-J) potential also describes the interaction between the fluid atom (polyethylene) and the wall atom (Ni):(11)$$E_{wf}\left(r\right)=4\varepsilon_{wf}\left[{\left(\frac{\sigma_{wf}}{r}\right)}^{12}-{\left(\frac{\sigma_{wf}}{r}\right)}^{6}\right]$$
where *w* and *f* denote wall and fluid atoms, respectively; *wf* represents the wall–fluid interaction; *E* is the interaction potential energy between wall atoms and fluid atoms; $\varepsilon_{\mathrm{wf}}$ and $\sigma_{\mathrm{wf}}$ are the characteristic energy and characteristic length of the LJ interaction potential, respectively. The definitions are as follows [24]:(12)$$\varepsilon_{wf}=\sqrt{\varepsilon_{w}\varepsilon_{f}}$$
(13)$$\sigma_{wf}=\frac{1}{2}(\sigma_{w}+\sigma_{f})$$

Normalized units were adopted to increase the accuracy of the computations and to generalize the problem researched because the actual quantities involved in the simulation are very tiny (on the atomic scale). The Boltzmann constant, the characteristic length and energy in the non-bonding potential between atoms, and the mass of an atom in the simulated system were all provided in unit 1. These units may be used to generate all other units. Table 2 below shows the most regularly physical quantities in normalized units and their conversion to international units.

### 2.3. Model Construction

The three-dimensional polymer melt plate Poiseuille flow model shown in Figure 2 is composed of the upper and lower Ni metal plates, as well as the center polyethylene. The following is the major building procedure:(1)The EAM potential was utilized to construct the Ni wall surface. Meanwhile, the wall surface as a whole was rigid, and its atoms were fixed in place at the lattice point location and employed to neutralize wall atom interactions. The rigid wall surface not only increases simulation efficiency but also enhances the accuracy of the fluid flow in the channel at low shear rates;(2)The modeling of polyethylene molecular chains was constructed in moltemplate [25]. It contains 200 chains with a degree of polymerization of 130. Afterwards, the generated file was subjected to systemic relaxation in open-source code Large-scale Atomic/Molecular Massively Parallel Simulator (LAMMPS) to obtain a stable polyethylene in the initial state with a time step of $∆t=4{.6\;\times \;10}^{-4}\;\tau$;(3)The entire model was composed using the potential parameters stated above, and the systemic relaxation procedure was re-performed to make the system more balanced and homogenous, using the final created file in steps (1) and (2). It is vital that the simulation temperature computation can only compute atomic thermal motion. However, the generation of kinetic energy during a non-equilibrium molecular dynamics simulation devastated the temperature calculation due to the macroscopic motion generated during the simulation; the temperature calculation must be modified to remove the influence of macroscopic kinetic energy [26,27]. As a result, an impartial temperature control strategy based on velocity profiles was employed. The system’s final measurements were 26.5σ × 26.5σ × 30σin the X, Y, and Z directions. In order to more clearly show the morphology of different molecular chains and the stratification between molecular layers, we used the Open Visualization Tool to set different colors for the model.

### 2.4. Research Scheme

Under macroscopic conditions, the pressure drop in the flow channel usually drives the Poiseuille flow of the polymer melts. However, the Poiseuille flow is simulated in molecular dynamics simulations by introducing a driving force *F_g_* to each atom in the *+x* direction [28]. As a result, it is critical to understand the involvement of driving forces in the flow process. We selected four atomic driving forces *(*$F_{g}=4.0,\;6.0,\;8.0,\;10.0$) for single factor simulation analysis. The relevant physical quantity units and the corresponding relationship with the SI units have been explained in Table 2. For various atomic driving forces, the transport and slip properties of the polymer flow process, as well as molecular structural changes, were observed.

The simulation zone in our study was separated into a number of thin layers with a thickness of Δ*z* = 0.25σ that are parallel to the walls, with the simulation region boundaries set to the location of the lowermost atoms near the upper wall and the uppermost atoms near the lower wall. The physical parameters such as velocity distribution and density distribution were derived by time averaging after obtaining a steady Poiseuille flow under the selected forces.

Numerous studies have demonstrated that the intensity of the wall–fluid interactions (WFI) has a significant impact on the rheological characteristics of the polymer fluid as well as the degree of wall slip [12,13,26,29]. For two different materials, the WFI between them is different, and the fluidity is also different, which will affect the simulation results. By altering the strength of interaction between the wall material Ni and the polymer PE molecular chain, the influence of WFI on the flow process and the creation of slip were explored in this work. The $\varepsilon_{\mathrm{wf}}$ is a parameter representing the strength of WFL. In past research [24], the value of $\varepsilon_{\mathrm{wf}}$ is usually obtained by Equation (12), and $\varepsilon_{\mathrm{wf}}=10$ is considered to be a moderate value at this time. In order to comprehensively study the effect of WFI on the slip, we investigated it at two different magnitudes: weak wall–fluid interactions (WWFI, $\varepsilon_{\mathrm{wf}}=1–5$) and strong wall–fluid interactions (SWFI, $\varepsilon_{\mathrm{wf}}=10–50$) about the flow of polymer melts in nanochannels.

## 3. Results and Discussion

### 3.1. Influence of Atomic Driving Force

#### 3.1.1. Velocity and Density Distribution

The different velocity profiles are shown in Figure 3A for the selected forces, where WFI was set as $\varepsilon_{\mathrm{wf}}=10$. Where the scatter is the actual thin layer velocity, which appears to slip, so it is fitted using Equation (5). The range of the driving force we mainly study is less than 10, but in order to confirm the influence law of the driving force, we also study the driving force greater than 10 at larger intervals. The results reveal that as the driving force increases, the maximum velocity that can be reached improves gradually. At the same time, the velocity distribution shows typical pseudoplastic characteristics of the polymer melt. The slip velocity trend is also depicted in Figure 3B. In order to clarify the variation trend of the slip velocity with the driving force, a wider range of simulations was carried out for the driving force. It is similarly positively associated with the driving force, and these results are consistent with prior simulations [12,29]. There are diversifications in the non-Newtonian exponents corresponding to the fitted equations for different forces, which is interesting (in Table 3). In fact, the experimental determination of the velocity distribution for polyethylene slit flow exhibits this feature [17], which is consistent with our simulation results. It represents a certain change in the degree of pseudoplasticity of the polymer melt. In Section 3.1.3, we will demonstrate the explanation for this occurrence on a molecular level.

Previous research [9,30] has demonstrated that fluid density distribution is critical for understanding the flow mechanism and its associated slip consequences. Figure 4B depicts the density distributions under various driving forces. Near the wall, the fluid density shows a visible trend of oscillation and stratification before gradually decaying to a uniform condition. The relative position of the fluid density distribution near the wall region does not change significantly. It indicates that driving forces do not have an effect on the general state of the fluid density distribution. At the same time, as the driving force increases, the amplitude of the initial peak of the density distribution, which is generally referred to as the contact density layer, has no significant change. It is consistent with previous research results [12,31] that the driving force has no effect on the density distribution during polymer melt flow. As the shear rate increases, the polymer sticks to the wall more firmly and generates a tighter lubricating layer along the interface [12], which effectively promotes the flow process. Due to the presence of a tiny gap between the wall and the fluid (Figure 4A), the zero-density location is not the real wall–fluid contact position. The axis-Y represents the actual contact location between the fluid phase and the FCC lattice plane, and it is frequently treated as a reference plane for slip velocity computations.

#### 3.1.2. Dynamic Information of Flow Process

The diffusion coefficient is a physical quantity that indicates the diffusion capacity of a fluid (or solid) and is also a key indicator of fluid mass transfer and activity capacity in nanochannels [32]. When there is no concentration difference, the self-diffusion coefficient, as one of the diffusion coefficients, is used to define the diffusion behavior of a single substance on its own. The self-diffusion coefficient has the following mathematical expression:(14)$$D=\underset{t\to \infty }{\lim}\frac{1}{2dNt}\langle {\sum_{i=1}^{N}\left[r_{i}(t)-r_{i}(0)\right]}^{2}\rangle$$
where *d* is the system dimension, which in this instance is 3 because it is a three-dimensional system, *N* is the total number of fluid particles in the flow system, and the mathematical formula in curly brackets is the mean square displacement of the fluid particles at the start and end of the entire flow.

The fluctuation of the self-diffusion coefficient under various driving forces is shown in Figure 5A. The self-diffusion coefficient rises with driving force, indicating that the driving force enhances the transport performance of the polymer melt during the Poiseuille flow. This trend can be observed in each time stage of flow, which is in line with the research conclusions of previous articles [12,13,14]. The fluctuation of the average velocity and the maximum speed that can be achieved at various forces is shown in Figure 5B. As the driving force rises, the average and maximum velocities steadily increase, which is consistent with the trend of the self-diffusion coefficient. It demonstrates how driving forces have a direct impact on polymer melt flow.

#### 3.1.3. Evolution of Internal Molecular Chain Structure

The previous section introduced the apparent polymer melt flow information as well as certain macroscopic characteristics. However, the origins of polymer melt behavior during flow must be determined at the molecular level (mainly polymer chain segment alterations), which is not only an advantage of molecular dynamics but also a requirement for understanding polymer melt flow processes at the nanoscale [11,33].

Entanglement is a measure that describes the presence of mutual restriction between adjacent strands. The entanglement parameter of the polymer melt chain is defined by the method mentioned by Yashiro et al. [34] and has the following expressions:(15)$$\theta_{i}=\frac{e_{i+10}\cdot e_{i-10}}{\left|e_{i+10}\right|\left|e_{i-10}\right|}$$

Taking one atom *i* as the starting point and  i − 10  and i+10 as the finishing points, two vectors $e_{i+10}$ and $e_{i\;-\;10}$ are created for a specific polymer chain, and then $\theta_{i}$ is determined using the preceding equation. If the entanglement parameter is greater than the entanglement threshold ($\theta_{t}$), the polymer chain atoms are regarded to be unentangled, and if it is less than the entanglement threshold, they are considered to be entangled.

The variation of the entanglement parameters throughout the whole process of Poiseuille flow is depicted in Figure 6. It represents the proportion of entangled state atoms to the total number of atoms. When no driving forces are applied, the entanglement parameter $\theta_{i}$ is usually distributed in the range of 80° to 150°, according to the legend. The total distribution is biased towards the extremes of 180° and 0° due to the application of driving forces on the atoms with the passage of time. The entanglement parameter $\theta_{i}$ tends to 180°, indicating that the chain segment is not only disentangled but also fully “stretched”, which can be considered to be in the orientation state by definition; while $\theta_{i}$ tends to 0°, the molecular chain may be in a “folded orientation state” at this time, despite being in the entangled state. As a result of this process, the degree of reciprocal binding between molecular chains reduces, and the molecular chains gradually untwist, releasing chain segments to migrate. The disentanglement of molecular chains promotes the increase in the kinetic parameters of the Poiseuille flow process for polymer melts (in Figure 5).

Polymer chains have a distinctive collective anisotropy due to their long linear chain structure, and under the influence of a driving force field, the chain segments will be arranged in a certain direction with preferential orientation, establishing the molecular chain orientation state [35]. It is demonstrated that the development of the orientation state alters some macroscopic properties of the polymer melt, such as the product’s optical properties and the modulus and strength in the tensile direction. The second Legendre polynomial coefficient is calculated in conjunction with the preceding approaches in the literature [36,37] to characterize the molecular chain orientation of the polymer melt during the simulation process. These are the precise expressions:(16)$$p_{2}=\frac{3}{2}\langle \cos^{2}\theta_{i}\rangle -\frac{1}{2}$$
where:(17)$$\theta_{i}=\frac{e_{i}\cdot e_{x}}{\left|e_{i}\right|\left|e_{x}\right|},e_{i}=\frac{r_{i+1}\cdot r_{i-1}}{\left|r_{i+1}\right|\left|r_{i-1}\right|}$$
$e_{x}$ is the direction of driving atomic forces exerted, i.e., the direction of flow of the polymer melt, and $e_{i}$ is derived by computing the $r_{i+1}$, $r_{i-1}$ coordinate vectors for the nearest surrounding atoms of $r_{i}$. When the coefficient $p_{2}=-0.5$, the molecular chain is perpendicular to the flow direction; when the coefficient $p_{2}=1$, the molecular chain is parallel to the flow direction, indicating that it is in the oriented state.

Figure 7A depicts the change in polymer molecular chain orientation as the nanosystem flows in the direction $e_{x}=\langle \;1,0,0\;\rangle$. The figure shows that with molecular chain migration and driving force influences, a considerable number of chain segments that are in the vertical state at the start (around $p_{2}=-0.5$) gradually diminish. Simultaneously, the substantial increase in chain segments around $p_{2}=1$ demonstrates the orientation process of polymer molecular chains in the system flow direction. The model snapshot depicts this transformation more clearly and intuitively. It can be observed that there are obvious differences in orientation states in different longitudinal regions in Figure 7B. We observed molecular snapshots of the X-Y view at different positions (upper, center, and lower). Among them, “upper” and “lower” are thin polymer layers close to the upper and lower walls, and “center” is the most central layer of polymer. Interestingly, the apparent orientation state originates in the thin polymer area near the metal Ni wall surface, and the orientated chain segments do not emerge in the center of the polymer melt from the beginning to the end in Figure 7C. On the one hand, the polymer adheres to the metal wall [12]; on the other hand, the wall–fluid interaction is assigned a value during the simulations near the upper and lower metal walls to the middle of the polymer melt where the wall–fluid interaction is gradually weakening. The non-equilibrium wall–fluid interaction, as well as the driving force making a joint impact along the flow direction, combine to cause the thin polymer layers closest to the wall to constitute the orientated state primarily. From near the upper and lower walls to the middle of the polymer, the range of orientation states of the thin polymer layers gradually expand.

Meanwhile, while comparing the orientation state under various driving forces, it is discovered that the degree of orientation of the polymer melt varies dramatically as the driving force enhances. The whole polymer melt molecular chains are entangled with each other in the initial state, and there is no discernible orientation. With increasing driving force, a larger proportion of the thin polymer layers reach the oriented state in Figure 8. Thin layers of the oriented state form and expand near the upper and lower metal walls, forming a parabolic-like velocity distribution, while the unoriented part (near the center of the polymer melt) remains entangled as a “polymer cluster”, which flows forward at the same velocity, forming a plunger flow. It is a non-Newtonian fluid with a distinctive attribute (pseudoplasticity). The thickness of the oriented thin layer increases as the driving force improves. The relative share of the parabolic-like velocity distribution and the plunger flow velocity distribution is changeable. The n-value(non-Newtonian exponent) varies as a result of this in Table 3. Larger driving forces facilitate higher flow velocities and slip velocities, but the velocity distribution eventually approaches the parabolic type of Newtonian fluids, i.e., the plunger flow component declines. It explains why the degree of pseudoplasticity of the polymer melt alters with changing driving force (Section 3.1.1).

The radius of gyration of polymer molecular chains exhibits the spatial structure of molecular chain segments, i.e., the spreading degree of molecular chains, in addition to the size of linear molecular chains. The radius of gyration can be calculated using the formula below [38]:(18)$$R_{g}{}^{2}=\frac{1}{n}{\sum_{i=1}^{n}\left(r_{i}-r_{cm}\right)}^{2}$$
where *n* represents the number of monomers on a single molecular chain, $r_{i}$ represents the position vector of monomers on the polymer molecular chain, and $r_{\mathrm{cm}}$ represents the position vector of the molecular chain center of mass. Because both the driving forces acting and the orientation direction are *x*-positive, the tensor *R_g_^2^-xx* is more representational. In this case, ${(r}_{i,x}\;{-\;r}_{\mathrm{cm},x})$ is used to replace ${(r}_{i}\;{-\;r}_{\mathrm{cm}})$. 

In order to verify that the larger driving force leads to an increase in the thickness of the oriented thin layer, we divide the fluid into three parts of equal thickness along the z-axis. From top to bottom are the upper surface layer, the core layer, and the lower surface layer. The tensor *R_g_^2^-xx* of the core layer and the surface layer were calculated, respectively. The results showed that the radius of gyration of the core layer decreases slightly with the increase in the driving force, while the radius of gyration of the surface layer increases greatly with the increase in the driving force. It is consistent with the findings in Figure 8, and Figure 9B describes the variation in the radius of gyration of the entire polyethylene melt. As the driving force strengthens, the radius of gyration tensor *R_g_^2^-xx* increases, suggesting that the polymer molecular chain is stretched in the *x* direction. The length of the molecular chain, on the other hand, does not make a significant difference when driving forces are applied. That iss because the tensor *R_g_^2^-xx* evolves when the spatial morphology of the polymer molecular chains happens to transform, i.e., the deepening of orientation rather than the elongation of molecular chain length. The radius of gyration tensor *R_g_^2^-xx* escalates as the driving force improves, indicating a higher degree of orientation of the polymer melt, which is consistent with the previous conclusion that as the driving force improves, more thin layers are generated in the orientated state.

### 3.2. Influence of Wall–Fluid Interaction

#### 3.2.1. Slip Velocity and Density Distribution

The difference in the WFI at the various polymer melt sites mentioned above is a major factor in the steady development of the orientation state from near the wall to the center. In prior research [29,39,40,41], it was believed that as the WFI strengthened, the flow of the polymer melt in the nanochannel would be impeded and that both the maximum velocity achievable and the slip velocity would be intensely reduced. 

The velocity distribution at various WFI resembles that of the preceding section and will not be repeated. The fluctuation in the slip velocity at various WFI is the main topic of discussion. To comprehensively study the impact of WFI, we selected a moderate driving force $F_{g}=5.0$. As in prior research, the slip velocity decreases as the WFI reinforces at the scale of SWFI, and when $\varepsilon_{\mathrm{wf}}=50$, the slip velocity approaches zero, and the boundary slip phenomena vanish. In contrast to SWFI, the slip velocity changes in the reverse direction with strengthening WFI at the scale of WWFI, where the strengthened WFI facilitates the slip of the polymer melt (Figure 10A). 

Correspondingly, the density distribution at different scales of WFI also has many different characteristics in Figure 10B. At the scale of WWFI, the overall relative position of the density distribution is stable, and only the peak of the contact density changes. At the scale of SWFI, not only the contact density peak value is different, but also the relative position of the density distribution changes. The weaker the WFI, the closer the contact density layer is to the wall.

The slip velocity is proportional to the net transition rate of fluid atoms in the first fluid layer along the flow direction [42]. The net transition rate here is equal to the difference between the atom transition rate in the direction of flow and the transition rate in the opposite direction. The molecular thermal motion of the Ni atoms at the surface of the wall atoms forms a folded potential energy field, which consists of a sequence of regularly ordered potential energy peaks and valleys due to their FCC lattice structure. Through the intensity of WFI, this potential energy field impacts the thin layer of fluid atoms near the upper and lower solid walls, commonly referred to as the first fluid layer (position of the contact density) [40]. The fluid atoms overcome the energy barriers between the peaks and valleys of the potential energy field to make a randomly oriented transition (Figure 11A): from one potential energy valley to the next potential energy valley. However, due to the addition of driving forces, the chance of transiting in the flow direction is substantially higher than in other directions, and prior authors [43,44] produced a mathematical relational notation for this process:(19)$$v_{s}=2\lambda \frac{k_{B}TF_{+}}{hF_{0}}\exp\left(-\frac{\Delta E}{k_{B}T}\right)\sinh\left(\frac{\tilde{\tau }\lambda S}{2k_{B}T}\right)$$

At this point, *T* represents the absolute temperature of the system, *k_B_* is the Boltzmann constant, and *h* is the Planck constant. *F_+_* and *F*_0_ are the coordination functions of the activated and initial states, respectively. Along the flow direction, *S* is the effective area of the first fluid layer, *v_f_* is the speed of thermal motion of the atoms in the first fluid layer, ∆E is the energy barrier to be crossed by the transition of the fluid atoms, and λ  is the horizontal distance between two potential energy valleys. When *v_s_ < v_f_,* the slip will only appear due to thermal motion; otherwise, there will be no slip. Because *S* is substantially less than *k_B_T* for low shear rates, the hyperbolic sine function may be reduced to a linear function, and *F*_+_/*F*_0_ can be considered to be constant. The equation may be simplified to [42]:(20)$$v_{s}\propto \tilde{\tau }\lambda^{2}S\exp(-\frac{\Delta E}{k_{B}T})$$

It can be seen that the variation in the slip velocity is mainly due to the different atomic transition barriers. Moreover, the atomic transition barrier is mainly related to WFI, the relative position from the solid surface, and the solid lattice structure.

During this simulation, the solid lattice structure has been FCC; the variation in the slip velocity is mainly determined by the competition between WFI and the relative position of different thin layers of the polymer melt and the wall surface [26]. The density distribution of the thin layers of the polymer melt defines the relative location of the distinct thin layers of the polymer melt to the wall surface. The location of the density distribution relative to the solid wall surface, especially the first fluid layer atoms, does not change at the scale of SWFI (Figure 10B). The WFI strength is currently a major influencing factor of the fluid atomic transition energy barrier, and strengthening the WFI will improve the transition energy barrier of the fluid atoms along the flow direction (Figure 11C), which is expressed in the simulation process as a decrease in polymer melt slip velocity and flow energy. At the [45] scale of WWFI, the density distribution is positioned relative to the solid wall; notably, the contact density layer atoms migrate toward the solid wall, and the fluid atoms are more widely distributed as the WFI attenuates. Because the atomic transition energy barrier increases rapidly as the distance from the wall decreases [26], the first fluid layer distribution will dominate the atomic transition energy barrier at this moment. The energy barrier of atomic transition emerges to climb at this point when the WFI attenuates (Figure 11C), and the polymer melt slip velocity and flow performance decline during the simulation process. The slip velocity and flow performance are greatly lowered when the WFI is too strong or too weak, as can be shown.

#### 3.2.2. Dynamic information of Flow Process

The self-diffusion coefficient increases with the WFI at each stage of the polymer melt flow process at the scale of WWFI, as shown in the stacking diagram in Figure 12A. At the same time, combined with the variety in average velocity under different WFI in Figure 12B reflects this phenomenon: when the WFI strengthens, the flow diffusion of the polymer melt in the channel enhancing. According to the prevailing opinion, strengthening the WFI will restrain fluid flow and mass transfer. However, the relative position of density distribution changes, leading to the change in wall slip at this time and producing the opposite result, and wall slip escalates with WFI (Section 3.2.1). The increase in wall slip promotes the flow and mass transfer of polymer melt. It reflects the massive effect of wall slip on polymer melt Poiseuille flow. As WFI strengthens, the maximum velocity that polymer melts can attain decreases simultaneously. 

As SWFI improves, the self-diffusion coefficient, average velocity, and maximum velocity at each step of the flow process tend to decline due to a decrease in the slip velocity, as shown in Figure 12C,D. When WFI is too strong, the slip velocity is close to 0 at this time, which will also cause the three factors to drop sharply. It is consistent with previous research findings that the enhancement of WFI would inhibit the polymer melt flow.

#### 3.2.3. Evolution of Internal Molecular Chain Structure

Figure 13 depicts the changes in the polymer molecular chain disentanglement and orientation characteristics at different scales of WFI. According to the scatter curve, a number of molecular chains are disentangled and establish an orientation state at the start of the flow, and then the entanglement and orientation state progressively moving to a rather stable condition. The disentanglement parameters illustrate two completely different changing states, and the orientation parameters are also the same. At the scale of WWFI, with the strengthening of WFI, both display a downward trend; at the scale of SWFI, with the strengthening of WFI, both display an upward trend. We observe that the molecular chain disentanglement and orientation happen at the same time and that the disentanglement and orientation promote each other. Considering the previous fluctuation of the slip velocity under the conditions of strong and weak WFI, the reasons for the occurrence of this simulation result are not difficult to deduce.

The impact of the slip phenomenon is that near the top and lower metal walls, the polymer molecular chain velocity is no longer zero but instead flows ahead with the flow front (the middle part of the polymer melt). As a result, the relative velocity of the polymer flow front and edge regions (near the top and lower metal walls) will decrease, resulting in a more aggregated whole. The slip velocity increases as WWFI improves, the entire polymer tends to flow forward as a whole, and the relative motion between the polymer layers parallel to the upper and lower walls weakens. It is difficult to separate and establish the orientation states. As SWFI improves, the slip velocity decreases, the entire polymer tends to gradually separate during the flow process, and the relative motion between the polymer layers parallel to the upper and lower walls strengthens, which makes the polymer molecular chains simpler to dissociate entanglement and formation of orientation states.

Previous studies [12] have shown that WFI has little effect on the radius of gyration, according to the simulation results. In our study, the radius of gyration tensor *R_g_^2^-xx* gradually grows larger throughout the flow. At the same time, similar to the findings of the orientation and disentanglement parameters, it reveals two entirely opposing evolving patterns. When WFI is enhanced, it reduces at the scale of WWFI in Figure 14A, whereas it improves at the scale of SWFI in Figure 14B. In fact, the driving force and WFI have radically different impacts on the spreading of the polymer melt molecular chains during the process of polymer melt flow. The polymer chains gradually stretch under the operation of the driving force, as seen in Figure 7. 

In the case of WWFI, the driving force is the main influencing factor of polymer chain extension at this moment. The enhancement of WFI is equivalent to a progressive lessening of the driving force impact; hence tensor *R_g_^2^-xx* reduces. The effect of WFI and driving force on the stretching of the molecular chain is nearly comparable in strength in the case of SWFI. The effect of WFI makes the polymer molecular chain tend to stay near the wall, and it plays a fixed role in the polymer chain, which is more conducive to the dragging effect of driving force on the molecular chain, improving the stretching degree of the molecular chain and increasing the radius of gyration. When $\varepsilon_{\mathrm{wf}}=50$, the radius of gyration shrinks because the impact of WFI has already exceeded the driving force, and a considerable number of molecule chains have been totally adsorbed in situ; therefore, it is not conducive to molecular chain stretching.

## 4. Conclusions

Based on the united-atomic molecular dynamics model, the Poiseuille flow process of polyethylene in nanochannels was simulated in this research. The effects of driving force, strong and weak wall–fluid interaction on the flow process, and slip of polyethylene were studied, respectively. Under these flow boundary conditions, model information such as velocity distribution, slip velocity, and density distribution, as well as kinetic information such as molecular chain diffusion, maximum and average velocity, and molecular chain spatial structure parameters such as disentanglement, orientation, and radius of gyration were obtained. The internal causal link of model information, kinetic information [33], and molecular chain information in this process was explained. The velocity distribution with pseudoplastic characteristics of the polymer melt is obtained, and its molecular origin is revealed. The polymer Poiseuille flow in nanochannels was re-understood at the molecular level. Because most polymer melts are pseudoplastic fluids, so the model presented in this paper and the pseudoplastic velocity distribution, kinetic changes, and molecular chain spatial morphology changes are applicable to other polymers. The following are the main findings reached:(1)In the flow process, the velocity distribution of the polymer melt represents its pseudoplasticity—that is, a plunger flow domain in the middle and two parabolic-like domains in the upper and lower side of the nanochannels rather than a parabolic flow—and the simulation findings match the high-density polyethylene flow experiments. The difference in the relative proportion of parabolic-like and plunger sections about velocity distribution is the result of the different thicknesses of the oriented thin layer;(2)Increasing the driving force improves the flow of the polymer melt while simultaneously reinforcing the amount of slip in the polymer melt, which makes it easier for the molecular chains to diffuse. The disentanglement of molecular chains and the development of an orientated state along the direction of driving pressures create this process, which lessens the barrier to flow caused by molecular chain mutual restriction;(3)At different scales of WFI, the slip velocity displays two entirely opposing states. At the scale of WWFI, the slip velocity decreases as the WFI strengthens. At the scale of WWFI, the contact density layer moves toward the wall when the WFI weakens, enhancing the atomic transition energy barrier and decaying the net atomic transition rate, resulting in a decreased slip velocity;(4)The specificity of slip velocity variation at different scales of WFI impacts not only molecular chain diffusion throughout the entire flow process but also causes certain changes in the polymer molecular chain conformation. With the strengthening of WFI, the disentanglement, orientation, and radius of the gyration *R_g_^2^-xx* parameters have been elevated at the scale of WWFI; on the contrary, they all decay at the scale of SWFI.

## Figures and Tables

**Figure 1 polymers-14-03218-f001:**
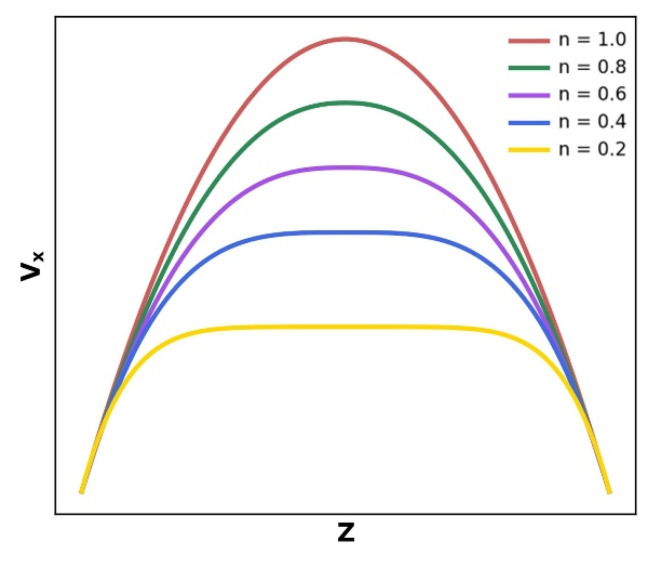
Velocity distribution at different non-Newtonian indexs (*n*).

**Figure 2 polymers-14-03218-f002:**
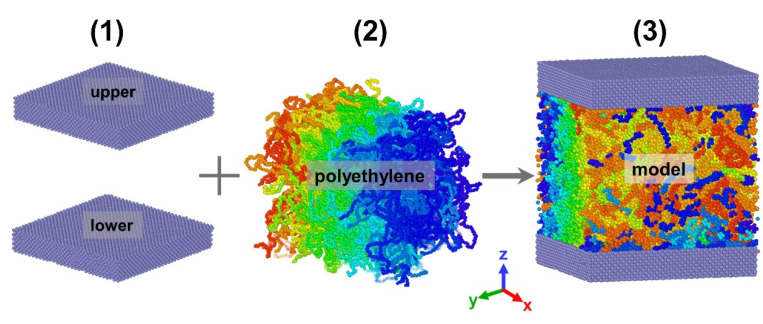
Construction process of Poiseuille flow model for polyethylene melt.

**Figure 3 polymers-14-03218-f003:**
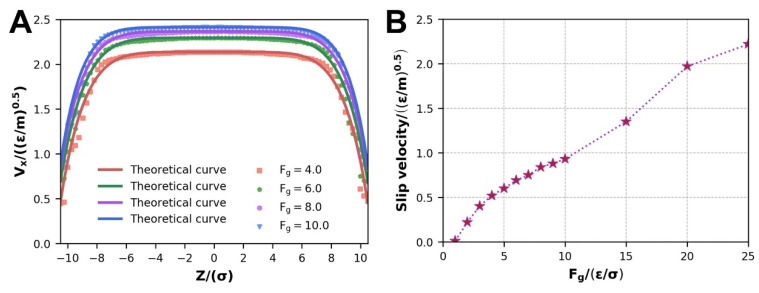
Under different driving forces: (**A**) velocity distribution and curve fitting of stable Poiseuille flow, (**B**) variation of slip velocity.

**Figure 4 polymers-14-03218-f004:**
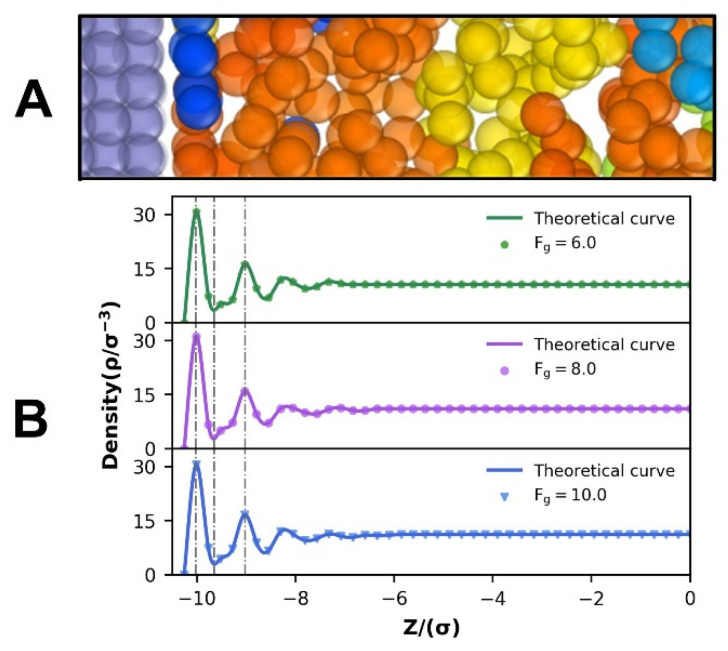
(**A**) Schematic illustration of the gap between the solid wall and the fluid contact section. (**B**) Density distribution of different driving forces.

**Figure 5 polymers-14-03218-f005:**
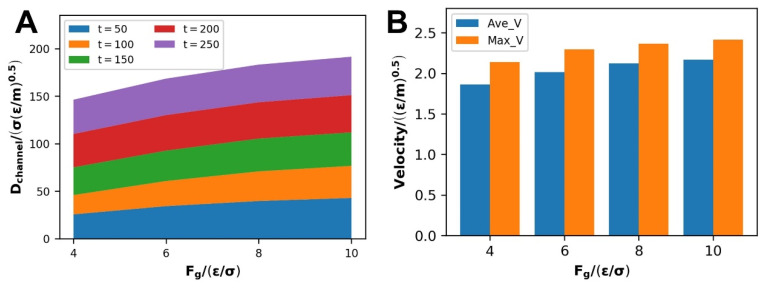
Under different driving forces: (**A**) stacked graph of self-diffusion coefficients at various stages, (**B**) average velocity and achievable maximum velocity during flow.

**Figure 6 polymers-14-03218-f006:**
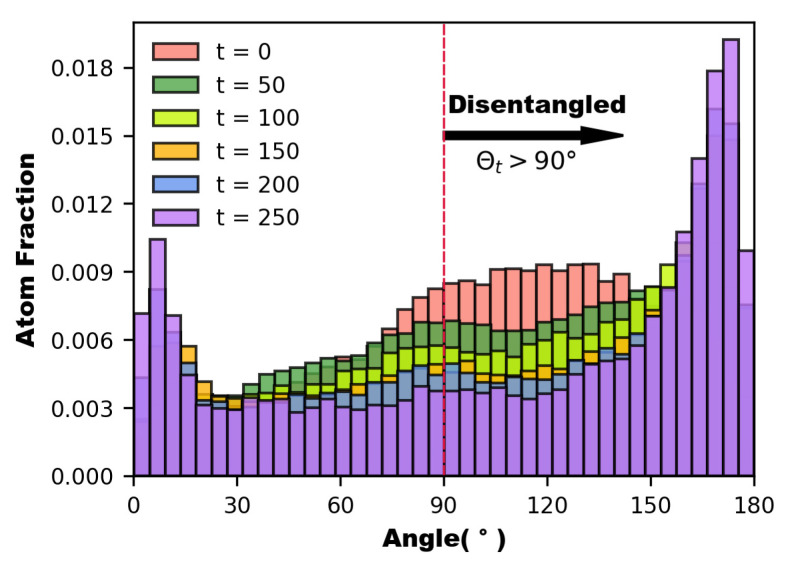
Statistical histograms of molecular chain entanglement and disentanglement at various time stages ($\theta_{t}=90{}^{\circ}$).

**Figure 7 polymers-14-03218-f007:**
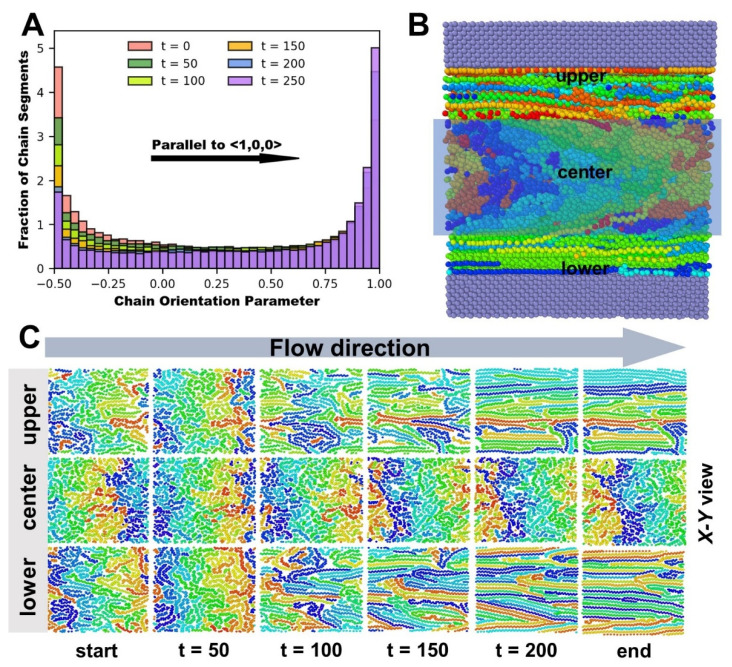
(**A**) Statistical histogram of molecular chain orientation parameters. (**B**) Molecular dynamics simulation last moment model snapshot. (**C**) Snapshots of X-Y cross-section molecular chains at different positions (upper, center, and lower) in polymer fluids at various time stages.

**Figure 8 polymers-14-03218-f008:**
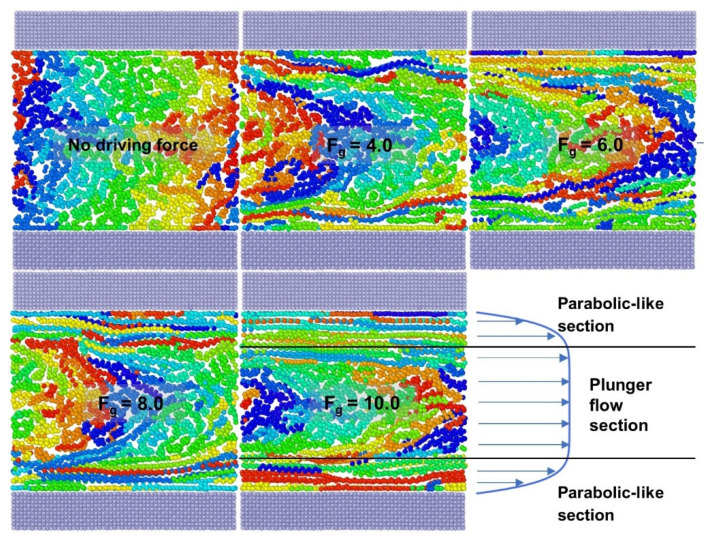
Comparison of molecular chain arrangement and velocity fitting curve of the model cross-section snapshot in the final stage of stable Poiseuille flow.

**Figure 9 polymers-14-03218-f009:**
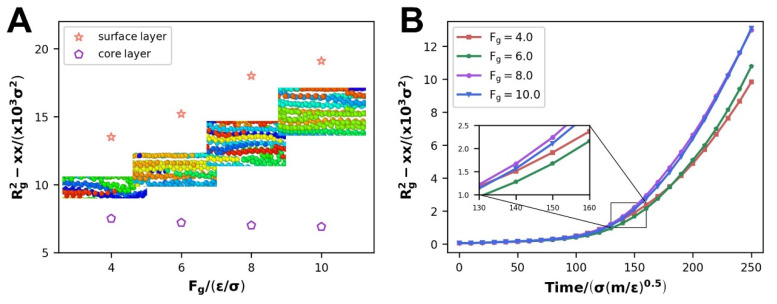
The change of the gyration tensor radius R_g_^2^-*xx* with the driving force: (**A**) the surface and core layers of the polymer melt, (**B**) the whole polymer melt.

**Figure 10 polymers-14-03218-f010:**
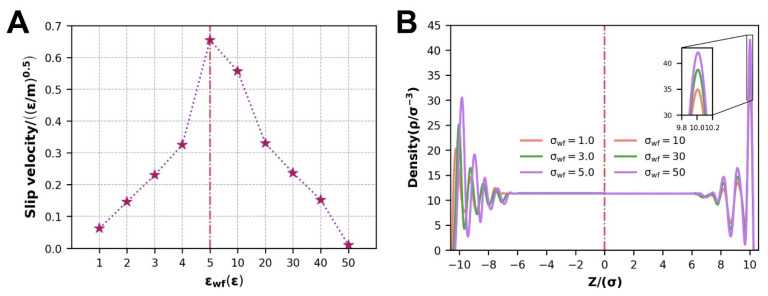
At different scales of WFI: (**A**) the point and line diagram of slip velocity, (**B**) density distribution.

**Figure 11 polymers-14-03218-f011:**
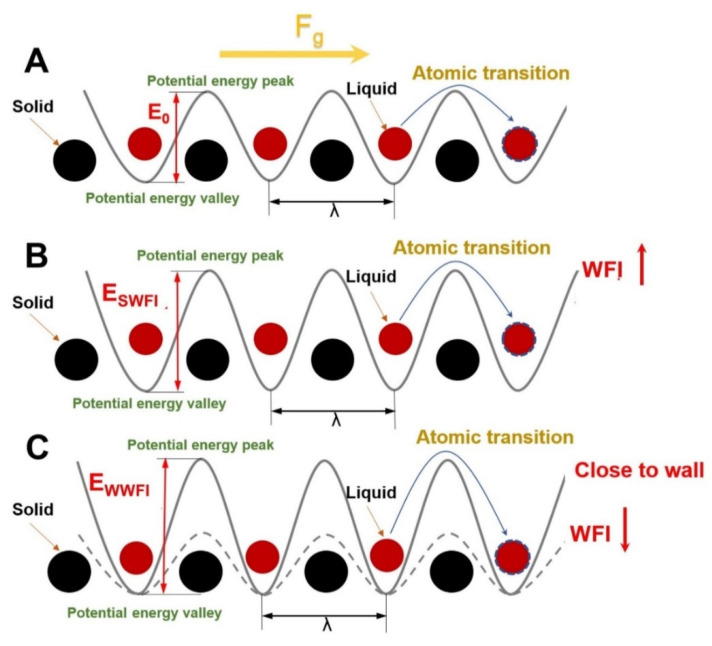
Schematic diagram of energy barrier and atomic transition: (**A**) at the scale of moderate WFI, (**B**) at the scale of SWFI, (**C**) at the scale of WWFI.

**Figure 12 polymers-14-03218-f012:**
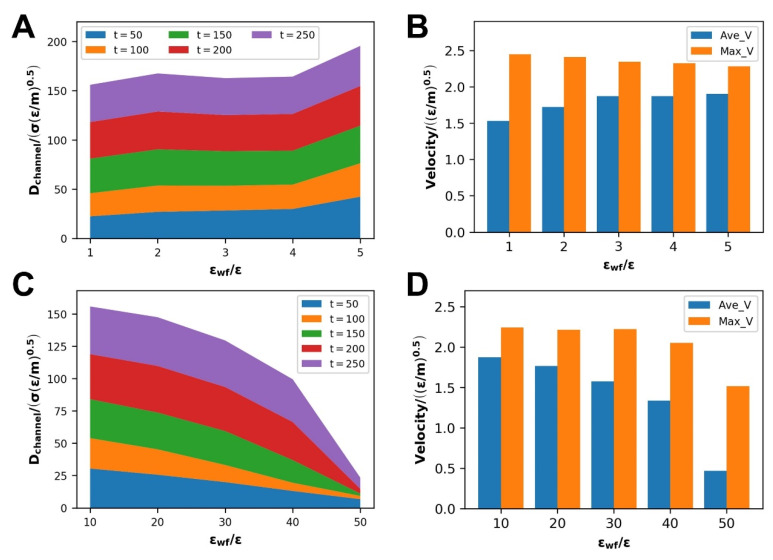
At the scale of WWFI: (**A**) stacked graph of self-diffusion coefficients at various stages, (**B**) average velocity and achievable maximum velocity during flow, at the scale of WWFI: (**C**) stacked graph of self-diffusion coefficients at various stages, (**D**) Average velocity and achievable maximum velocity during flow.

**Figure 13 polymers-14-03218-f013:**
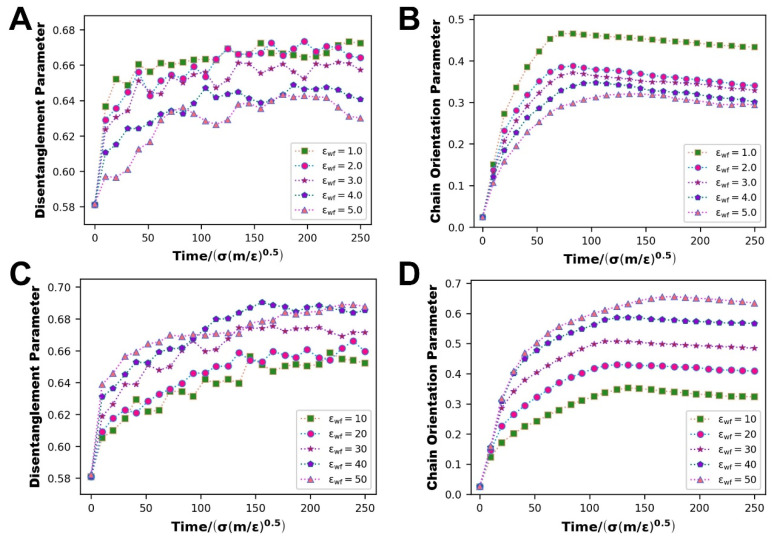
At the scale of WWFI: (**A**) molecular chain disentanglement parameters, (**B**) molecular chain orientation parameters, at the scale of SWFI: (**C**) molecular chain disentanglement parameters, (**D**) molecular chain orientation parameters.

**Figure 14 polymers-14-03218-f014:**
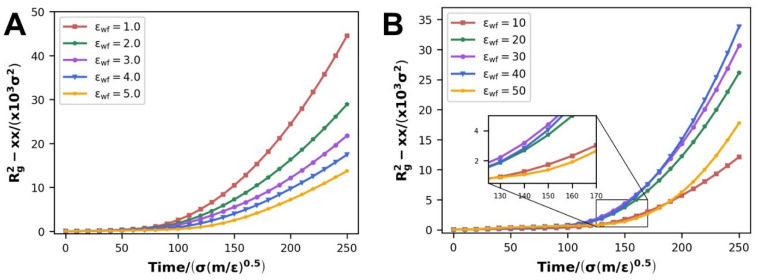
Radius of gyration tensor considering flow direction: (**A**) at the scale of WWFI, (**B**) at the scale of SWFI.

**Table 1 polymers-14-03218-t001:** Force field and potential parameters for PE in MD simulation.

Interaction	Type	Parameters
Bond length	X-CH_2_	$K_{b}=3125{.0\varepsilon ,\;r}_{0}=0.38\sigma$
Bond angle	X-CH_2_-CH_2_	$K_{\theta }=535{.86\;\varepsilon /\mathrm{rad}}^{2}{,\;\theta }_{0}=1.911\;\mathrm{rad}$
Dihedral angle	X-CH_2_-CH_2_-CH_2_	*C*_0_$=15.50,\;C_{1}=-39.97,\;C_{2}=6.93,\;C_{3}=62.42(\varepsilon$)
Non-bonded	CH_2_	$\varepsilon \;{=1,\;\sigma =1,\;r}_{c}=2.5\sigma$
	Ni	$\varepsilon_{\mathrm{Ni}}=106{.93\varepsilon ,\;\sigma }_{\mathrm{Ni}}=0{.57\sigma ,\;r}_{c}=2.5\sigma$

**Table 2 polymers-14-03218-t002:** Normalized unit of common physical quantities and conversion of SI units.

Physical Quantity	Normalized Unit	International Unit
Mass	*m*	14 g/mol
Distance	σ	$4{.01\;\times \;10}^{-10}\;m$
Energy	ε	$7{.78\;\times \;10}^{-22}\;J$
Time	$\tau =\sigma {\left(m/\varepsilon \right)}^{1/2}$	$2{.18\;\times \;10}^{-12}\;s$
Temperature	$T=\varepsilon /k_{B}$	56.52 K
Velocity	$v={\left(\varepsilon /m\right)}^{1/2}$	$1{.83\;\times \;10}^{2}\;m/s$
Driving force	F=ε/σ	$1{.95\;\times \;10}^{-12}\;N$

**Table 3 polymers-14-03218-t003:** Parameters of velocity profiles fitting under different driving forces.

F_g_	$v_{0}$	$v_{s}$	$v_{\max}$	n
4	1.676	0.46	2.136	0.15
6	1.605	0.69	2.295	0.17
8	1.526	0.84	2.366	0.18
10	1.533	0.88	2.413	0.20

## Data Availability

Data presented in this study are available on request from the first author.

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
