# Peer review of "Molecular Insights into the Wall Slip Behavior of Pseudoplastic Polymer Melt in Nanochannels during Micro Injection Molding"

_polymers, 2022, doi:10.3390/polym14153218_

Round 1

Reviewer 1 Report

The authors presented an effective modeling method for polyethylene (PE) melt flow in nanochannels based on united-atom molecular dynamics simulations. The slip velocity, dynamics information of the flow process, and spatial configuration parameters of molecular chains were characterized to study effects of driving forces and wall-fluid interactions on the behavior of PE melt under Poiseuille flow conditions. The results showed that the proposed model can describe the pseudo-plastic behavior in nanochannels better than other commonly used model. The work is interesting and can be published in Polymers if the following issues can be addressed:

1. Some chemical formulas are not properly presented (Ex. Page 4 line 119)

2. How was FCC of Ni chosen for this simulation? In practice, mold can be made by different type of materials. How do different materials affect the simulation results?

3. In Figure 3, why were more data for driving force below 10 but less for the others?

4. What are the meaning of different colors in Figures 2, 4A, 7B and C, 8, 9A? Axis label of most Figures should be improved with proper name.

5. In Figure 7C, at the same driving force, how could the length of molecular chain affect their alignment in the upper and lowers layer at the end of the simulation?

6. Can the proposed model be applicable to other polymers?

7. There are some grammar errors in the manuscript. Writing needs to be improved.

Reviewer 2 Report

The manuscript with the title of “Molecular insights into the wall slip behavior of pseudoplastic polymer melt in nanochannels during micro injection molding” has been well written. The novelty, description and results are proper. The authors used the active verb in some sentences. Also, the number of references in too much. After, modifying these two issues, it can be accepted.
